# Don't Judge a Book by its Cover - On the Dynamics of Recurrent Neural Networks

## Abstract

To be effective in sequential data processing, Recurrent Neural Networks (RNNs) are required to keep track of past events by creating *memories*. Consequently RNNs are harder to train than their feedforward counterparts, prompting the developments of both dedicated units such as LSTM and GRU and of a handful of training tricks. In this paper, we investigate the effect of different training protocols on the representation of memories in RNN. While reaching similar performance for different protocols, RNNs are shown to exhibit substantial differences in their ability to generalize for unforeseen tasks or conditions. We analyze the dynamics of the network's hidden state, and uncover the reasons for this difference. Each memory is found to be associated with a nearly steady state of the dynamics whose speed predicts performance on unforeseen tasks and which we refer to as a 'slow point'. By tracing the formation of the slow points we are able to understand the origin of differences between training protocols. Our results show that multiple solutions to the same task exist but may rely on different dynamical mechanisms, and that training protocols can bias the choice of such solutions in an interpretable way.

## 1 Introduction

Recurrent Neural Networks (RNN) are the key tool currently used in machine learning when dealing with sequential data (Sutskever et al., 2014), and in many tasks requiring a memory of past events (Oh et al., 2016). This is due to the dependency of the network on its past states, and through them on the entire input history. This ability comes with a cost - RNNs are known to be hard to train (Pascanu et al., 2013a). This difficulty is commonly associated with the vanishing gradient that appears when trying to propagate errors over long times (Hochreiter, 1998), and has been addressed by introducing various architectures such as Long Short-Term Memory (LSTM) cells (Hochreiter & Schmidhuber, 1997), Gated Recurrent Units (Chung et al., 2015).

In addition to network architecture, training protocols can also facilitate training success. Inspired by shaping in behavioral training, curriculum learning (Bengio et al., 2009; Cirik et al., 2016) suggests gradually increasing task demands until arriving at the full task. This was shown to accelerate training, albeit reaching fairly similar results after convergence of training (Jozefowicz et al., 2015; Cirik et al., 2016).

For complex tasks, however, there are many possibilities on how to gradually increase difficulty. In this work, we ask whether different curricula indeed lead to the same outcome. We choose a task in which there is a natural division of computational resources that is common for many RNN tasks - memory and processing. RNNs are not only required to memorize the input, but are also trained to process it. This processing, which could perhaps be easier for a feed-forward network, utilizes the same connections that are trained for the memory component of the task. We study a simple task combining these elements, which fails for naive training.

We thus start training the network on one aspect of the task; either first learning to extract features or first learning to memorize, and when successful moving to learn the second aspect as well. Both approaches work, leading to comparable performance on the full task. A more careful analysis, however, reveals that the resulting networks differ in their extrapolation abilities and reflect their training histories.

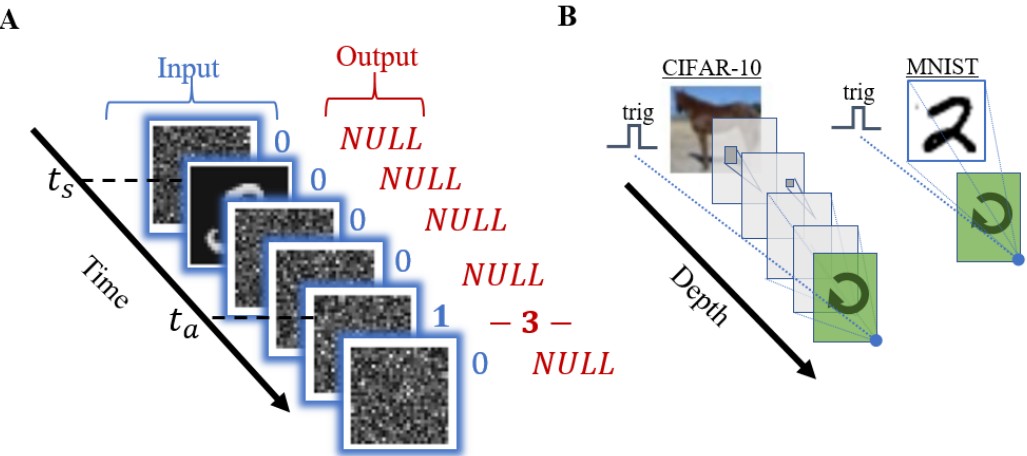

Figure 1: **A** Task. The network is presented an MNIST or CIFAR-10 image amidst noisy images, and has to report its label at a later time, as requested by a separate input ($0, 1$ to the right of images). Output should be null at all times except the reporting time. The precise times $t_a, t_s$ vary from trial to trial. **B** Architecture. In case of MNIST dataset, both image and the trigger signal are fed directly into the recurrent layer. For CIFAR-10 task a convolutional feed forward network is added in front of the recurrent layer, while trigger signal is kept connected directly to the RNN.

Understanding the underlying cause for these differences requires some form of reverse engineering. This can be done by focusing on individual recurrent units (Karpathy et al., 2015; Oh et al., 2016), or by analyzing global network properties. We opt for the latter, analyzing the RNN's hidden states as a discrete-time dynamical system.

In this framework, memories might be associated with a wide range of dynamical objects. On one extreme, transient dynamics can be harnessed for memory operations Manjunath & Jaeger (2013); Maass (2011); Maass et al. (2002). On the other extreme, there are memory networks Sukhbaatar et al. (2015) that memorize *everything* and later use only the relevant memories while ignoring all the rest. The idealized dynamical scenario where each memory is associated with a fixed point in the RNN state space (Hopfield, 1982; Sussillo, 2014; Barak, 2017; Amit, 1989) was refined in (Sussillo & Barak, 2013; Mante et al., 2013) where points which are only approximately fixed (slow points), with a drift that is slower than the task duration, were shown to represent memory.

To this end, we extend tools used in continuous-time systems in neuroscience (Sussillo & Barak, 2013) to address the aforementioned differences in extrapolation abilities. Examining relations between drift speed of memories, which we find to be represented by slow points of varying slowness, and the extrapolation properties of their associated class, we establish such a correlation in a large variety of settings.

Finally, we obtain an instructive insight on *how* the memory robustness is altered along the training course. Detailed analysis of individual training trajectories makes it possible to monitor the formation of slow points under a specific training protocol. This technique offers us an explanation of the interplay between newly created and existing slow points – decreasing stability of the latter in a systematic and tractable manner. This provides a link between training curriculum, dynamical objects, memory and performance.

## 2 TASK DEFINITION

Inspired by real-world applications of Recurrent Neural Networks (Oh et al., 2016), we designed a task where the RNN has to combine stimulus processing and memorization (figure 1). The network is presented with a series of noisy images, among which appears a single target image (from MNIST or CIFAR) at time $t_s$. At a later time point, $t_a$, the network receives a response trigger in a separate

input channel, prompting it to output the label of the image. At all other times, the network should report a null label.

The stimulus and reporting times are chosen randomly each trial from a uniform distribution on $[1, T_{max}]$ subject to the constraint $t_a - t_s > 4$. The total stimulation time $T_{max} = 20$, and the network was requested to distinguish between $|V| = 10$ different classes of MNIST (LeCun et al., 2010) or CIFAR-10 (Krizhevsky et al.).

Each pixel of the noise mask was sampled from a Gaussian distribution with mean and variance matching its counterpart at the image corpus $\epsilon \sim N(\mu_n, \sigma_n^2)$. We tested the RNNs ability to extrapolate from this task in two directions, the first was increasing the delay $T_{max}$, and the second was increasing the noise $\sigma^2$.

The motivation behind this task is three-fold. First, as explained, this task is comparable to real-world scenarios where RNNs are used for, combining the need for both stimulus memorization and feature extraction. Second, the task lends itself to parametric variations, allowing to compare both different training protocols and generalization abilities. Third, desiring to understand the dynamical nature of memorization in discrete Gated-RNNs, the delay between stimulus and response trigger allows for evolution of RNN hidden-state (HS), which can be reliably analyzed using well known methods from dynamical systems (Sussillo & Barak, 2013), which we modify to our discrete setting.

## 3 MODEL

For MNIST, the network consists of a single recurrent layer of $d = 200$ gated recurrent units, an output layer of $|V| + 1 = 11$ neurons, $|V| = 10$ neuron for the different classes, and an additional neuron for the null indicator. The input layer has $n + 1$ neurons, where $n$ is the number of pixels in the image and an extra binary input channel for the response trigger $X_r(t)$ defined by:

$$X_r(t) = \begin{cases} 1, & \text{if } t = t_a. \\ 0, & \text{otherwise.} \end{cases} \tag{1}$$

For CIFAR-10, the network was expanded to $d = 400$ recurrent units, along with a convolutional front-end composed of three convolutional layers and two dense layer. To eliminate issues of translational invariance regarding the response trigger and the convolutional front-end, the trigger was added as an extra channel to the final dense layer, right before the recurrent units.

The gated units are either GRU or LSTM:

$$
\begin{aligned}
z &= \sigma(W_z I + U_z h_t + b_z) \\
r &= \sigma(W_r I + U_r h_t + b_r) \\
h_{t+1} &= \tanh(W_{h_t} I + U_h(r \circ h_t) + b_h)
\end{aligned}
\tag{2}
$$

$$
\begin{aligned}
f &= \sigma(W_f I + U_f h_t + b_z) \\
i &= \sigma(W_i I + U_i h_t + b_i) \\
o &= \sigma(W_o I + U_o h_t + b_o) \\
c_{t+1} &= f \circ c_t + i \circ \tanh(W_c I + U_c h_t + b_c) \\
h_{t+1} &= o \circ \tanh(c_t)
\end{aligned}
\tag{3}
$$

For the analysis of the network's phase space, we denote the state of the recurrent layer by $\xi$, which for LSTM is $\xi = \begin{pmatrix} h \\ \tanh(c) \end{pmatrix}$ and for GRU $\xi = h$.

The network was trained using the 'Adam' optimizer (Kingma & Ba, 2014) with a soft-max cross-entropy loss function with a increased loss on reporting at $t = t_a$ in proportion to $T_{max}$. Full description of each protocol, including schedules and other hyper-parameters is given in Appendix C.

## 4 TRAINING PROTOCOL: TWO TYPES OF CURRICULA

We found that training failed when using straightforward SG optimization on the full task. The network converged to a state where it consistently reports 'null' without regarding neither the output trigger nor the images it has received as inputs. This suboptimal behavior did not improve upon further training. On the other hand, we observed that simpler versions of the task are learn-able.

If the maximal delay between stimulus and reporting time was short or when we introduced only a limited number of different digits, the network was able to perform the task. This led us to try two different protocols of curriculum learning in order teach the network the full required task:

1. Vocabulary curriculum (VoCu) - here we started from two classes $'V' = \{c_1, c_2\}$ and then increased the vocabulary gradually until reaching the full class capacity. This protocol is similar to the original concept of (Bengio et al., 2009) except the fact that in our vocabulary all the classes occur with the same frequency, and the selected order of curricula is in fact arbitrary.

2. Delay curriculum (DeCu) - starting from short delays between stimulus and reporting time ($T_{max} = 6$), we progressively extended it toward the desired values. Implicitly mentioned in (Hochreiter, 1998), this regime is expected to mitigate the vanishing gradient problem, at least during initial phase of training.

## 5 EXTRAPOLATION ABILITY DEPENDS ON TRAINING PROTOCOL

We found that, in good accordance with existing literature (Bengio et al., 2009; Jozefowicz et al., 2015) results for the nominal test-set were fairly indifferent to the training protocol (Appendix A). Once we evaluated the ability of each setting to extrapolate to more challenging task settings, however, similarity ends and differences emerge.

We tested two different extrapolation settings, motivated by different hypotheses on the underlying mechanism of the network. As mentioned in the Introduction, prior work provides a wide spectrum of alternative hypotheses – from systems without stable states, through slow points, and to robust fixed points.

We evaluated these hypotheses by either extending the delay or adding noise. First, we observed how each setting performs when the delay between stimulus and response trigger is extended further beyond $T_{max} = 20$. If $|V| = 10$ robust fixed-point attractors have formed, retrieval accuracy should not be affected by the growing delay. If the computation is based on transients, then all class information is expected to eventually vanish.

Experiments revealed that neither of these exterme options was the case - performance deteriorated with increasing delay, but did not reach chance levels (figure 2). This deterioration implies that not every memorized digit corresponds to a stable fixed point attractor, but some do. Furthermore, the deterioration was curriculum-dependent, with DeCu outperforming VoCu for all cases except CIFAR-GRU.

Similarly, we can evaluate reporting accuracy when a the noise mask has a higher variance than used for training [1]. Once more, this serves to evaluate the hypothetical fixed points, and figure 2 shows that not all curricula are equally able to mitigate input-driven noise which can indicates differences in the basin-of-attraction of the hypothetical attractors formed under each setting.

Both extrapolation experiments show that despite comparable results superficially, the governing dynamics and representation of the networks vary with training protocol, resulting in different extrapolation performance. An additional extrapolation experiment requiring reporting the class twice with two triggers also shows large differences between the protocols (Appendix B).

Both extrapolation experiments reveal a similar trend, with the DeCu protocol being superior to VoCu in three out of the four conditions studied. For the CIFAR-10 dataset with a GRU architecture, we found that DeCu extrapolates worse than DeCu. We will return to this difference in the next section.

## 6 DYNAMICS OF HIDDEN REPRESENTATION

The relevant phase space of this dynamical system is the recurrent layer state $\xi$. We thus begin by visually inspecting (in the first 3 PCA components) the activity of the network for the maximal training delay, $\Delta t = 20$. We show here results for the MNIST dataset with a GRU architecture, but

---

[1]Since Gaussian noise is not translational invariant, For CIFAR-10, zero mean noise was added directly to the input of the recurrent layer, instead of increasing the input-noise variance.

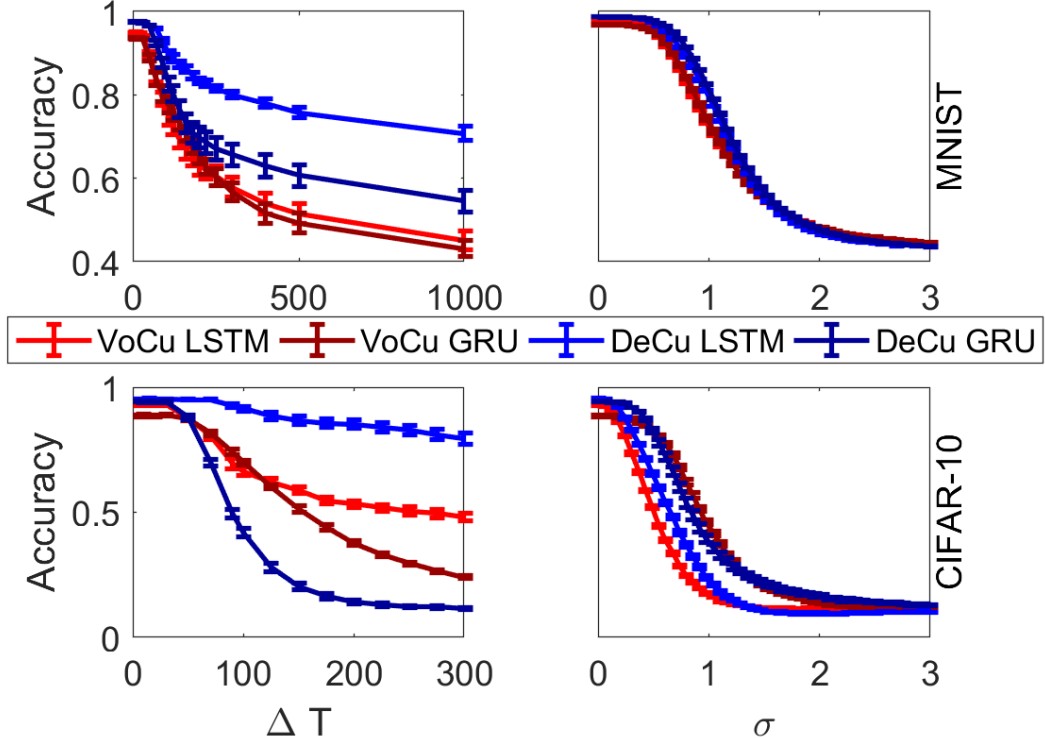

Figure 2: Retrieval accuracy when increasing the delay between stimulus and response trigger beyond $T_{max} = 20$ (*left*) and for increasing noise standard deviation beyond $\sigma_n$ (*right*) For MNIST and CIFAR-10 (inset). Despite similar performance initially, the ability to generalize for greater delays and greater noises than trained for varies with protocol. On the MNIST dataset, DeCu was superior in both LSTM and GRU architectures compared to VoCu, while with CIFAR-10, results on LSTM were qualitatively similar to the results on MNIST. However, with the GRU architecture on the CIFAR-10 dataset, VoCu's performance was preferable to Decu's.

similar behavior is seen for other conditions and the statistics of all conditions is analyzed below. The left panels of Figure 3 show that different trials of each digit are well separated into $|V|$ regions with a one to one correspondence to data classes. Following these trajectories for a longer delay of $\Delta t = 1000$ shows that some regions converge into what appears to be fixed points, while others vanish (middle panels). This is true even when a smaller noise amplitude is used (right panels), although here more regions survive. These figures also clearly show the difference between the two protocols. While both achieve a good separation with the nominal delay (left), it is already apparent that VoCu leads to clouds of points with a larger spread, possibly indicating a weaker attraction.

To verify the existence or absence of fixed points hinted by the above visualization, we apply an algorithm developed for continuous time vanilla RNN Sussillo & Barak (2013) to our setting. Briefly, fixed points (stable or unstable) are local minima of the (scalar) speed $S(\xi, I)$ of the hidden state $\xi$.

$$S(\xi, I) = ||F(\xi, I) - \xi||_2 \tag{4}$$

where the evolution of state,

$$F(\xi') = \xi(t+1)\Big|_{\xi(t)=\xi'} \tag{5}$$

is given by equations equation 2 or equation 3 for GRU or LSTM respectively. It is now possible to use gradient descent on the speed $S$ with respect to state $\xi$, namely, $\nabla_\xi S$, to locate such minima.

The initial conditions for this gradient descent were obtained by running the network with the mean delay value $\Delta t = 15$, and using the average of hidden states of each class. The external input

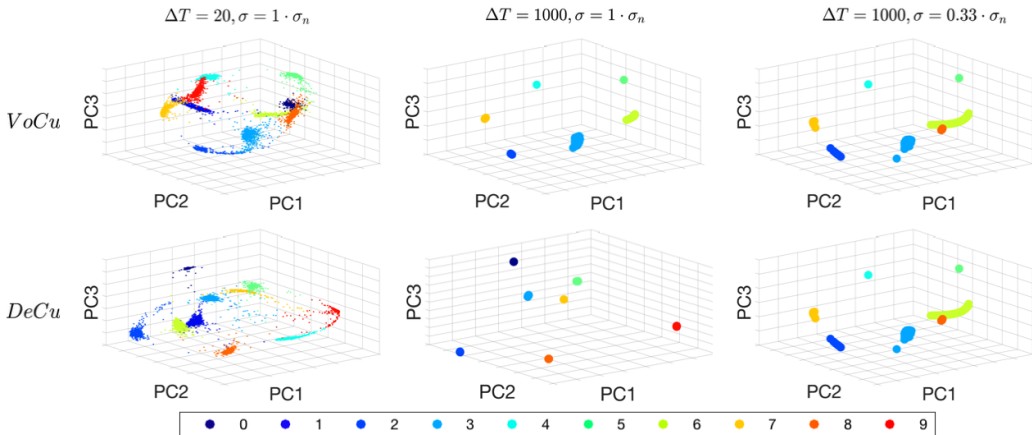

Figure 3: Hidden state projected on leading principal components in GRU - RNN on MNIST. Delays of $\Delta t = 20$ and $10^3$ time-steps are depicted with latter shown for different input noise strength. States are color codded by their label in the dataset. For the nominal delay ten distinct regions are observed in the state space corresponding to each of the $|V| = 10$ classes. While this picture is in agreement with the naive hypothesis that each class has its dedicated fixed-point, examination of a larger delay $\Delta t = 10^3$ reveals that while some classes collapse into a single point others vanish completely. Testing whether this disappearance due to insufficient basin of attraction around true fixed points or because of points being slow rather completely fixed we observe that also under small noises, there is still an evanescence of certain classes for large delays, albeit more classes survive. The Spread of samples along with fewer number of distinct fixed points at $\Delta t = 10^3$ in VoCu align with our findings of faster and less stable dynamics compared to DeCu discussed in sequel.

$I$ during gradient descent was the average of the noise images, thus effectively making our system Time-Invariant where such points and their stability are well defined. We verified that using different fixed external inputs did not qualitatively alter the results (not shown). We repeated the procedure for several realizations, and it always resulted in a local minimum of speed for each class (which we call slow point), with a readout that matches the class label.

Our hypothesis was that slow points with higher speed would be predictive of low extrapolation performance, and thereby explain the variability of Fig 2. We thus located slow points for every class of several network realizations for all settings. For every slow point, we computed its speed, and the accuracy of its associated class after a long delay. Figure 4 shows that the speed of the slow point associated with a certain class can predict how members of that class will react to extrapolation experiments. This trend holds for all architectures, unit types and datasets tested.

The colored bars in Figure 4 denote the mean and standard deviation (not standard error) of the speeds obtained by the two different protocols. The picture here is consistent with that observed in Figure 2, with DeCu outperforming VoCu for three of the four cases. Our results suggest that this difference is mediated by a difference in speed of the associated slow points. What exactly is the reason for the different behavior of the CIFAR dataset with a GRU architecture is left for future work.

## 7 FORMATION OF SLOW POINTS - WHY CURRICULA DIFFER

We saw that the two training protocols lead to a different representation of the stimulus memory by the network, and to different dynamical objects. How does training give rise to these differences? To answer this question, we analyze in detail one setting - a GRU architecture trained on the MNIST database. We follow the slow points of the velocity backwards in training time to learn how they arise and change throughout training, and correlate these events with network performance.

---

[2] For clarity, only five out of the ten classes are shown

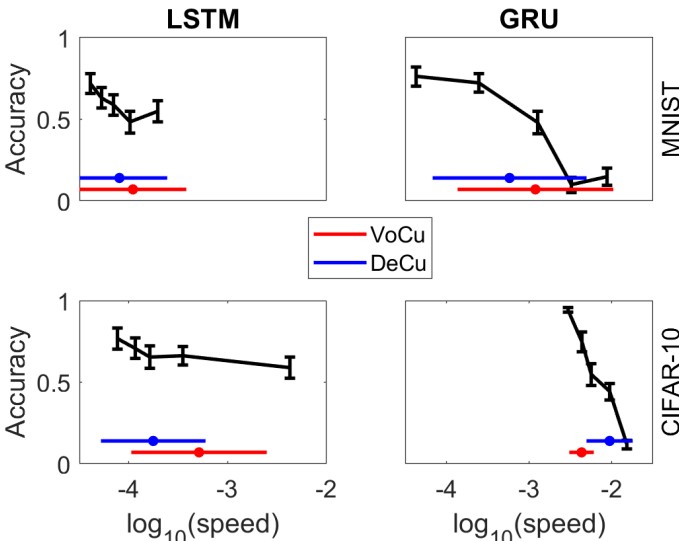

Figure 4: We measured the speed of slow points, and computed the extrapolation accuracy of the associated class for long delays. In all datasets, unit-types and training methods, slower speeds correlate with increased accuracy. Errorbars in the black curves denoted standard error of the mean. The colored bars show the mean and standard deviation of the speeds for each training protocol. Except VoCu-LSTM ($p = 0.07$) all differences are significant ($p < 0.05$). The difference in speeds between the protocols underlies the different extrapolation performance shown in Figure 2. Ten networks were used for MNIST, and five for CIFAR-10.

We located slow points as described in section 6, and then used them as initial conditions for gradient descent on the network defined by the previous training step. We then repeated this procedure iteratively to all training steps. The assumption is that the change in network parameters at each training step will not induce a very large shift in the locations of the relevant slow points, and thus our continuation procedure can track them. This is not clear in the case of VoCu, where one might expect discontinuities whenever a new class is added. Looking at the speeds of the tracked slow points shows that this is indeed the case for VoCu (figure 5, **A,B**), with all tracked speeds exhibiting such jumps. DeCu, on the other hand, shows a gradual slowing down of the slow points throughout training.

By observing the gradual slowing down in DeCu, it is quite easy to understand why this protocol improves performance - slow points become slower. But the situation for VoCu is more complicated. A natural expectation might be that the classes that were presented first will have more time to stabilize, and thus will be the slowest. We saw that this is not the case (not shown), and thus proceeded to look deeper into VoCu. We inspected the jumps in VoCu more carefully, knowing that this is where the tracking algorithm could fail.

We thus perform a short tracking procedure for each class - starting before the succeeding class is introduced, and following it back to the first training iteration in which it appeared. The result is a training-trajectory of a slow point, from the moment it first appears until the next class is introduced. For each point in this trajectory, we calculated the readout of the network. By stitching together the results of all short-tracking procedure of all classes, we obtain a full bifurcation diagram (5 **C**). Our results show that slow points do not emerge out of thin air, but rather bifurcate from existing slow points. For example class '7' emerged from classes '3' or '4'.

Such an event might affect the stability of classes '3' and '4', prompting us to evaluate this possibility. To more accurately observe the extent of such effect, we add noise to reduce network performance, and then check the difference in performance of the various classes following the introduction of class '7'. Figure 5 (**D**) shows that indeed the robustness of classes '3' and '4' was adversely affected compared to the other classes. To evaluate the statistics of this phenomenon, we repeat this procedure for many networks, and all class introduction events. Figure 5 (**E**) com-

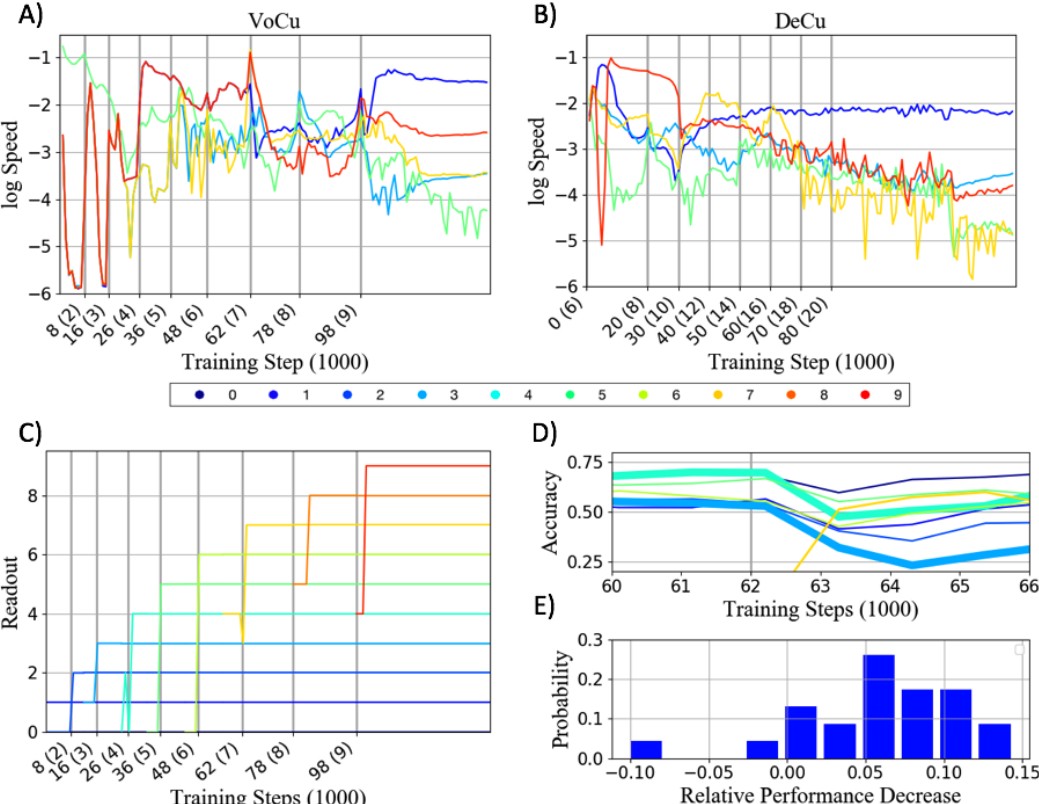

Figure 5: **A,B** Speeds of each slow-point through training [2] obtained by iteratively tracking them back in training time. The specific schedule of each curriculum is marked on the time axis. VoCu shows sharp jumps in the speed of all points for each class introduction, in contrast to DeCu which exhibits a gradual slowing down along the whole course of training. **C** A bifurcation diagram for VoCu, obtain from short backtracking every class proximal to its appearance. The change in readout suggests which slow point gave rise to the new point (e.g., 8 from 5 at time 78; 7 from 3 or 4 at time 62). **D** Under twice the nominal noise the accuracies of each individual class for the same VoCu realization when class '7' was introduced. As a result, all previously existing classes show a degradation in accuracy, however, the classes from which class '7' emerged from (classes '3' and '4') exhibited a stronger decline. **E** Verifying the statistical significance of the result in (D). For each bifurcation event, we comparing the accuracy drop of classes that gave rise to the new class, with the drop for the remaining classes. The histogram is from all events in three VoCu realizations, and shows that indeed accuracy of spawning classes decreases more following a bifurcation.

pares the change in accuracy of the classes that spawned the new class with the decay in accuracy of the other classes. We thus see that adding new classes corresponds to adding new slow points, which emerge from existing ones. The performance of the spawning classes is specifically adversely affected by these events.

# 8 DISCUSSION

Training RNNs to perform tasks is difficult (Pascanu et al., 2013b), and as a consequence many suggestions were made on how to alleviate this difficulty. Changing network architecture or unit-types might be expected to generate different solutions to the same task. Using different training protocols, however, is thought to allow better convergence to similar solutions.

Here we showed that different training protocols can lead to different locally optimal solutions. Although these solutions perform similarly under nominal conditions, challenging the networks with unforeseen settings reveals their differences.

An RNN is a dynamical system, and as such its operation can be understood in the language of fixed points and other dynamical objects. By analyzing the phase space of the network's hidden states, we showed that the memory of each class was associated with a slow point of the dynamics. The speeds of these slow points were highly correlated to the functional characteristics of memory longevity, and thus provide a dynamical explanation of the idiosyncrasies observed between curricula and architectures. Our result proved valid across architectures, datasets and unit types.

For one example, we were able to follow the formation of the aforementioned slow points during the training process. We characterized the topology of the phase space and showed that slow-points bifurcate one form another upon introduction of new classes. This bifurcation event is associated with a decrease in network accuracy that is specific for the classes involved in the bifurcation. Things could have been different - slow points could emerge in an area of phase space that is distant from existing ones, and the introduction of a new class could have resulted in a uniform effect on all existing classes.

To uncover this bifurcation structure, we introduced a backtracking methodology that could be relevant to any case in which learning modifies the dynamics of the network. Possible applications include studying the success and failure in creating memories, preventing catastrophic forgetting, understanding memory capacity and more.

Slowly increasing the delay resulted in better stability and increased extrapolation ability for three out of the four conditions tested. For the CIFAR-10 database with GRU units, we observed the opposite trend. While we did not uncover the reason for this anomaly, we speculate that once the issue is understood and resolved it will outperform VoCu. We note that even in this setting the same strong negative correlation between slow-point speed and memory still holds.

The setting studied in this work is a particular case of the more general problem of solution equivalence in gradient based optimization: On the one hand theoretical and numerical evidence does exist for training outcome's indifference to protocol details. Specifically to our case of RNN, it is shown in (Cirik et al., 2016) that, at least for language modeling tasks, performance does not heavily depend on training protocol. On the other hand, such an indifference is far from being fully established. In particular, stochastic gradient optimization suffers from known drawbacks (Dauphin et al., 2014; Martens & Sutskever, 2011) and might prove dependent on initialization (Sutskever et al., 2013) (and in particular pre-training). Our results show that networks with the same initialization can reach different solutions, and begin to uncover the dynamics underlying the route to these solutions.

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

# A Accuracies on Test and Train sets for nominal task

Below are the accuracies on the MNIST and CIFAR-10 train and test sets for the nominal task, for both GRU and LSTM architectures and the described curricula.

### Table 1: MNIST - GRU

|              |        | DeCu             | VoCu             | Naive |
|--------------|--------|------------------|------------------|-------|
| Training Set | Null   | 100%             | 100%             | 100%  |
|              | Digits | $97.5 \pm 0.2\%$ | $94.3 \pm 0.8\%$ | 0%    |
| Test Set     | Null   | 100%             | 100%             | 100%  |
|              | Digits | $96.5 \pm 0.3\%$ | $93.8 \pm 0.7\%$ | 0%    |

### Table 2: MNIST - LSTM

|              |        | DeCu             | VoCu             | Naive |
|--------------|--------|------------------|------------------|-------|
| Training Set | Null   | 100%             | 100%             | 100%  |
|              | Digits | $97.3 \pm 0.1\%$ | $95.2 \pm 0.3\%$ | 0%    |
| Test Set     | Null   | 100%             | 100%             | 100%  |
|              | Digits | $97.3 \pm 0.1\%$ | $95.2 \pm 0.2\%$ | 0%    |

### Table 3: CIFAR - GRU

|              |        | DeCu             | VoCu             | Naive |
|--------------|--------|------------------|------------------|-------|
| Training Set | Null   | 100%             | 100%             | 100%  |
|              | Digits | $94.3 \pm 0.9\%$ | $88.8 \pm 2.0\%$ | 0%    |
| Test Set     | Null   | 100%             | 100%             | 100%  |
|              | Digits | $72.4 \pm 1.0\%$ | $65.7 \pm 1.2\%$ | 0%    |

### Table 4: CIFAR - LSTM

|              |        | DeCu             | VoCu             | Naive |
|--------------|--------|------------------|------------------|-------|
| Training Set | Null   | 100%             | 100%             | 100%  |
|              | Digits | $95.1 \pm 0.4\%$ | $92.9 \pm 0.8\%$ | 0%    |
| Test Set     | Null   | 100%             | 100%             | 100%  |
|              | Digits | $66.5 \pm 0.4\%$ | $67.7 \pm 1.2\%$ | 0%    |

# B Accuracies when requiring a second retrieval

Accuracies when extending the nominal task to two trigger setting. Specifically, after the first trigger is introduced, we add another one and observe the decay in accuracy upon that second trigger.

Table 5: Second Retrieval MNIST

|  |  | DeCu | VoCu |
|---|---|---|---|
| GRU | First trigger at $t = 12$ | $97.6 \pm 0.1\%$ | $93.6 \pm 0.6\%$ |
|  | Second trigger at $t = 12$ | $97.2 \pm 0.1\%$ | $79.8 \pm 5.4\%$ |
|  | Difference | $0.4 \pm 0.2\%$ | $13.9 \pm 5.3\%$ |
| LSTM | First trigger at $t = 12$ | $97.4 \pm 0.1\%$ | $94.9 \pm 0.3\%$ |
|  | Second trigger at $t = 12$ | $97.3 \pm 0.2\%$ | $86.3 \pm 5.1\%$ |
|  | Difference | $0.2 \pm 0.3\%$ | $8.7 \pm 5.1\%$ |

Table 6: Second Retrieval - CIFAR

|  |  | DeCu | VoCu |
|---|---|---|---|
| GRU | First trigger at $t = 12$ | $93.2 \pm 0.2\%$ | $89.7 \pm 0.4\%$ |
|  | Second trigger at $t = 12$ | $93.2 \pm 0.3\%$ | $85.9 \pm 0.3\%$ |
|  | Difference | $0.1 \pm 0.3\%$ | $3.8 \pm 3.2\%$ |
| LSTM | First trigger at $t = 12$ | $95.7 \pm 0.1\%$ | $93.2 \pm 0.1\%$ |
|  | Second trigger at $t = 12$ | $95.7 \pm 0.3\%$ | $92.91 \pm 0.3\%$ |
|  | Difference | $0.03 \pm 0.4\%$ | $0.3 \pm 0.3\%$ |

## C  HYPER-PARAMETERS

### C.1  MNIST

The network was trained for a total of $14 \cdot 10^4$ Gradient Decent steps, using the ADAM optimizers. For the first $12 \cdot 10^4$ steps, the learning rate was $10^{-4}$ and for the latter steps the learning rate was decreased by a factor of 10. The DeCu schedules was starting with a maximal delay of $T_{max} = 6$, then incrementally increasing the maximal delay by two timesteps at the following training steps: $[20, 30, 40, 50, 60, 70, 80] \cdot 10^3$. For VoCu, training started by learning the first two classes, and each class what added in order according at these training steps: $[8, 16, 26, 36, 48, 62, 78, 98] \cdot 10^3$. The noise mask for this dataset was a normal distribution with mean $\mu = 0.1307$ and standard deviation $\sigma = 0.30816$.

### C.2  CIFAR-10

For CIFAR-10, a convolutional front-end was added consisting for three convolutional layers with 64 each, and an additional two fully-connected layers with 1024 neurons each. The network was for $20 \cdot 10^4$ Gradient Decent steps, with an initial learning rate of $10^{-4}$ and was decreased by a factor of 10 for the final $2 \cdot 10^{-4}$ steps. The DeCu scheduling was at steps $[30, 45, 60, 75, 90, 105, 120] \cdot 10^3$ and the VoCu scheduling was $[12, 24, 38, 54, 72, 92, 124, 140] \cdot 10^3$. The noise mask for this dataset was a normal distribution with mean $\mu = 0.4734$ and standard deviation $\sigma = 0.2517$.

