# OpenReview forum: "DON’T JUDGE A BOOK BY ITS COVER - ON THE DYNAMICS OF RECURRENT NEURAL NETWORKS"
_ICLR.cc/2019/Conference_

### Official Review · AnonReviewer3 · 2018-10-14
**An intriguing but preliminary investigation into RNN dynamics and generalisation**

**Rating:** 6
**Confidence:** 3

**Review:**

Post-rebuttal update:
The authors have clarified their main messages, and the paper is now less vague about what is being investigated and the conclusions of the experiments. The same experimental setup has been extended to use CIFAR-10 as an additional, more realistic dataset, the use of potentially more powerful LSTMs as well as GRUs, and several runs to have more statistically significant results - which addresses my main concerns with this paper originally (I would have liked to see a different experimental setup as well to see how generalisable these findings are, but the current level is satisfying). Indeed, these different settings have turned up a bit of an anomaly with the GRU on CIFAR-10, which the authors claim that they will leave for future work, but I would very much like to see addressed in the final version of this paper. In addition some of the later analysis has only been applied under one setting, and it would make sense to replicate this for the other settings (extra results would have to fit into the supplementary material).

I did spot one typo on page 4 - "exterme", but overall the paper is also better written, which helps a lot. I commend the authors on their work revising this paper and will be upgrading my rating to accept.

---

The authors investigate the hidden state dynamics of RNNs trained on a single task that mixes (but clearly separates) pattern recognition and memorisation. The authors then introduce two curricula specific to the task, and study how the trained RNNs behave under different deviations from the training protocol (generalisation). They show that under the curriculum that exhibited the best generalisation, there exist more robust (persisting for long time periods) fixed/slow points in the hidden state dynamics. They then extend the optimisation procedure developed by Sussillo & Barak for continuous-time RNNs in order to find these points. Finally, they use this method to track the speed of these points during the course of training, and link spikes in speed to one of the curricula which introduces new classes over time.

Understanding RNNs - and in particular how they might "generalise" - is an important topic of research. As done previously, studying RNNs as dynamical systems is a principled way to do so. In this line of work some natural objects to look into are fixed points and even slow points (Sussillo & Barak) - how long they can persist, and how large the basins of attraction are. While I believe the authors did a reasonable job following this through, I have some concerns about the experimental setup. Firstly, only one task is used - based on object classification with images - so it is unclear how generalisable these findings are, given that the authors' setup could be extended to cover at least another task, or at least another dataset. MNIST is a sanity check, and many ideas may fail to hold when extended to slightly more challenging datasets like CIFAR-10.

Secondly, as far as I can tell, the results are analysed on one network per setting, so it is hard to tell how significant the differences are. While some analyses may only make sense for single networks, e.g. Figure 3, a proper quantification of some of the results over several training runs would be appropriate.

Finally, it is worth investigating LSTMs on this task. This is not merely because they are more commonly used than GRUs, but they are strictly more powerful - see "On the Practical Computational Power of Finite Precision RNNs for Language Recognition", published at ACL 2018. Given the results in this paper and actually the paper that first introduces the forget gate for LSTMs, it seems that performing these experiments solely with GRUs might lead to wrong conclusions about RNNs in general.

There are also more minor spelling and grammatical errors throughout the text that should be addressed. For example, there is a typo on the task definition on page 2 - "the network should *output* a null label."

---

> ### Author Response · Authors · 2018-11-27
> **Response to Reviewer 3**
>
> (New version uploaded, including improved clarity, CIFAR-10, LSTM, new bifurcation analysis)
>
> We thank the reviewer for the detailed review and suggestions.
>
> Following the comments from all reviewers, we have clarified our main messages.
> Our main results are:
> 1.1) It is known and we also show: Different curricula can lead to the same performance.
> 1.2) Despite that, when extrapolating the task to new settings, differences between networks trained with these curricula arise.
> 2) The source of these differences can be traced to dynamical objects formed through training - in our case slow points.
> 3) Analyzing these slow points in the nominal task can predict behavior on the extrapolated task
> 4) Tracing the formation of these slow points through learning provides a link between training, slow point formation, stability of memories and performance
>
> Answers to specific comments below: (C - reviewer comment, R - response)
>
> C1) Firstly, only one task is used - based on object classification with images - so it is unclear how generalisable these findings are, given that the authors' setup could be extended to cover at least another task, or at least another dataset. MNIST is a sanity check, and many ideas may fail to hold when extended to slightly more challenging datasets like CIFAR-10.
>
> R1) We extended the setup to cover another dataset: CIFAR-10.
>
> C2) Secondly, as far as I can tell, the results are analysed on one network per setting, so it is hard to tell how significant the differences are. While some analyses may only make sense for single networks, e.g. Figure 3, a proper quantification of some of the results over several training runs would be appropriate.
>
> R2) We repeated the analysis for 20 realizations of MNIST (10 with GRU and 10 with LSTM), and 10 realizations of CIFAR-10 (5 GRU, 5 LSTM). All figures now include error bars.
>
>
> C3) Finally, it is worth investigating LSTMs on this task. This is not merely because they are more commonly used than GRUs, but they are strictly more powerful - see "On the Practical Computational Power of Finite Precision RNNs for Language Recognition", published at ACL 2018. Given the results in this paper and actually the paper that first introduces the forget gate for LSTMs, it seems that performing these experiments solely with GRUs might lead to wrong conclusions about RNNs in general.
>
> R3) We repeated all the experiments on LSTM, finding qualitatively similar phenomena.
>
> C4) There are also more minor spelling and grammatical errors throughout the text that should be addressed. For example, there is a typo on the task definition on page 2 - "the network should *output* a null label."
> R4) We fixed this and many other typos and grammatical errors.

---

> ### Author Response · Authors · 2018-12-05
> **Response to Reviewer 3: post-rebuttal comments**
>
> We thank the referee for a prompt response and constructive comments.
>
> Regarding the anomaly revealed with the GRU on CIFAR-10: further investigation of this case shows that increasing regularization leads to DeCu outperforming VoCu, as in all other scenarios.
>
> As the referee requested, we will present more bifurcation portraits in the final submission.

---

### Official Review · AnonReviewer1 · 2018-11-01
**Learned memory structure differs due to training paradigm**

**Rating:** 7
**Confidence:** 4

**Review:**

This paper titled <don't judge a book by its cover - on the dynamics of recurrent neural networks> studies how different curriculum learning results in different hidden state dynamics and impacts extrapolation capabilities. By training a 200-GRU-RNN to report the class label of a MNIST frame hidden among noisy frames, authors found different training paradigms resulted in different stability in memory structures quantified as stable fixed points. Their main finding is that training by slowly increasing the time delay between stimulus and recall creates more stable fixed point based memory for the classes.

Although the paper was clearly written in a rush, I enjoyed reading it for the most part. These are very interesting empirical findings, and I can't wait to see how well it generalizes.

# I find the title not very informative. Connection from 'Book' to 'Curriculum' is weak.

# The task does not have inherent structure that requires stable fixed points to solve. In fact, since it only requires maximum 19 time frames, it could come up with weird strategies. Since the GRU-RNN is highly flexible, there would be many solutions. The particular strategy that was learned depends on the initial network and training strategy.

# How repeatable were these findings? I do not see any error bars in Fig 2 nor table 1.

# How sensitive is this to the initial conditions? If you use the VoCu trained network as initial condition for a DeCu training, does it tighten the sloppy memory structure and make it more stable?

# I liked the Fig 2b manipulation to inject noise into the hidden states.

# English can be improved in many places.

# Algorithm 1 is not really a pseudo-code. I think it might be better to just describe it in words. This format is unnecessarily confusing and hard to understand.

# Does the backtracking fixed/slow point algorithm assume that the location of the fixed point does not change through training? Wouldn't it make more sense to investigate the pack-projection of desired output at each training step?

# PTMT, PMTP, and TaCu are not described well in the main text.

# The pharse 'basin of attraction' is losely used in a couple of places. If there isn't an attractor, its basin doesn't make sense.

# Fig 4 is not very informative. Also is this just from one network each?

# Fig 5 is too small!

# page 2: input a null label -> output a null label

# it would be interesting to see how general those findings are on other tasks, e.g., n-back task with MNIST.

---

> ### Author Response · Authors · 2018-11-27
> **Response to Reviewer 1 - Part 1**
>
> (New version uploaded, including improved clarity, CIFAR-10, LSTM, new bifurcation analysis)
>
> We thank the reviewer for the positive review, as well as for the detailed comments and suggestions.
>
> Following the comments from all reviewers, we have clarified our main messages.
> Our main results are:
> 1.1) It is known and we also show: Different curricula can lead to the same performance.
> 1.2) Despite that, when extrapolating the task to new settings, differences between networks trained with these curricula arise.
> 2) The source of these differences can be traced to dynamical objects formed through training - in our case slow points.
> 3) Analyzing these slow points in the nominal task can predict behavior on the extrapolated task
> 4) Tracing the formation of these slow points through learning provides a link between training, slow point formation, stability of memories and performance
>
> Answers to specific comments below: (C - reviewer comment, R - response)
>
> C1) I find the title not very informative. Connection from 'Book' to 'Curriculum' is weak.
> R1) Our main point was that although different protocols lead to the same performance in the nominal settings, their internal dynamics “under the hood” are different - hence the proverb.
>
>
> C2) The task does not have inherent structure that requires stable fixed points to solve. In fact, since it only requires maximum 19 time frames, it could come up with weird strategies. Since the GRU-RNN is highly flexible, there would be many solutions. The particular strategy that was learned depends on the initial network and training strategy.
> R2) Indeed for short delays, transients can suffice. But the variable delay is expected to encourage a fixed point solution (Orhan and Ma, bioRxiv 2018). In the revised version, we also explicitly state several possible strategies. Our result shows that indeed different training strategies can lead to different solutions.
>
> C3) How repeatable were these findings? I do not see any error bars in Fig 2 nor table 1.
> R3) We repeated the analysis for 20 realizations of MNIST (10 with GRU and 10 with LSTM), and 10 realizations of CIFAR-10 (5 GRU, 5 LSTM). All figures now include error bars.
>
> C4) How sensitive is this to the initial conditions? If you use the VoCu trained network as initial condition for a DeCu training, does it tighten the sloppy memory structure and make it more stable?
> R4) Training for VoCu and successively for DeCu is similar to letting VoCu more training time at the final stage with all classes introduced. As we discussed in the main text that additional training does not enhance performance. Regarding sensitivity to initial conditions, we performed our analysis for several initialisations of each setting and all results between settings were alike.
>
> C5) I liked the Fig 2b manipulation to inject noise into the hidden states.
> R5) Thanks!
>
> C6) English can be improved in many places.
> R6) We edited the text, and hope that the English has been improved.
>
> C7) Algorithm 1 is not really a pseudo-code. I think it might be better to just describe it in words. This format is unnecessarily confusing and hard to understand.
> R7) We described the algorithm (and the new short back-tracking algorithm) in words.

---

> ### Author Response · Authors · 2018-11-27
> **Response to Reviewer 1 - Part 2**
>
> C8) Does the backtracking fixed/slow point algorithm assume that the location of the fixed point does not change through training?
> R8) It assumes that it doesn’t change a lot. This is a reasonable assumption in numerical continuation - which is what we do here. We also state this assumption explicitly in the text now. This assumption can break down near bifurcations, which is exactly what happens in VoCu - and we now have a new part in the text that analyzes these bifurcations, linking them to changes in performance.
>
> C9) Wouldn't it make more sense to investigate the pack-projection of desired output at each training step?
> R9) The new short backtracking algorithm combines pack-projection with backtracking. After learning a new class, it makes more sense to do the gradient descent at the relevant training step. When we want to study the emergence of a slow point, however, backtracking is needed. In the new part of the text dealing with bifurcations, we show how this method can reveal which existing slow points give rise to new ones when a class is learned.
>
> C10) PTMT, PMTP, and TaCu are not described well in the main text.
> R10) We removed these protocols, as they did not contribute much to the main message.
>
> C11) The pharse 'basin of attraction' is losely used in a couple of places. If there isn't an attractor, its basin doesn't make sense.
> R11) A slow point can also have a region of attraction, albeit in a shape of saddle rather than a basin . Eventually, the dynamics will drift away from this slow point - but there is an area of phase space that will initially lead to the slow point. We removed or modified places where this term was used inappropriately.
>
> C12) Fig 4 is not very informative. Also is this just from one network each?
> R12) We replaced this with a more informative figure showing the dependence of accuracy on the speed of the relevant slow point, rather than just the speed itself.
>
> C13) Fig 5 is too small!
> R13) It was enlarged, and new panels were added to address the bifurcation analysis, revealing the root cause of glitches in speed depicted in the original submission..
>
> C14) page 2: input a null label -> output a null label
> R14) fixed.
>
> C15) it would be interesting to see how general those findings are on other tasks, e.g., n-back task with MNIST.
> R15) We thank the reviewer for proposing this task. Since some of key aspects of this study -  in particular, and crucially, time generalization for unforeseen delays are not easily extendable to this case , we decided to leave this for future work.

---

### Official Review · AnonReviewer2 · 2018-11-03
**understanding memorization vs processing across two types of curricula in RNNs**

**Rating:** 5
**Confidence:** 4

**Review:**

This manuscript attempts to use a delayed classification task to understand the dynamics of RNNs.  The hope is to use this paradigm to distinguish memorization from processing in RNNs, and to further probe when specific curricula (VoCu, DeCu) outperform each other, and what can be inferred from that.

Quality:
- The experimental design is sensible.  However, it is rather too much a toy example, and too narrow, hence it is unclear how much these results can be generalized across RNNs
- Highly problematic is that the key concepts in the paper -- memorization and processing -- are not well defined.  This means that the results inevitably are just interpretations rather than any sort of compelling empiricism.  After a careful read of the paper, I found it difficult to take away any particular learnings, other than "training RNNs is hard."

Clarity:
- The paper is fairly straightforward, which is positive.
- The lack of clarity around particular definitions means that clarity is limited to the empirical results.  If the results are incredibly compelling, that would be acceptable, but absent that (as is the case here), the paper comes across to me as rather unclear in its purpose or its takeaway message.

Originality:
- The Barak 2013 paper seems to be the key foundation for this work.  This work is sufficiently original beyond that paper.

Significance:
- The combination of lack of clarity and limited results on a toy setting imply that the significance is rather too low.

Overall, this is a genuine effort to explore the dynamics of RNNs.  I suggest improvements can be made by either (1) working hard to clarify in the text *exactly* what question is being asked and answered, or (2) broadening the results to make a much more rigorously supported point, or (3) ideally both.

---

> ### Author Response · Authors · 2018-11-27
> **Response to Reviewer 2**
>
> (New version uploaded, including improved clarity, CIFAR-10, LSTM, new bifurcation analysis)
>
> We thank the reviewer for the comments, and apologize for the lack of clarity in the previous version.
>
> Briefly, the distinction between memorization and processing was part of the motivation, and was used to construct the curricula. It was not, however, the main result.
> Following the comments from all reviewers, we have clarified our main messages.
> Our main results are:
> 1.1) It is known and we also show: Different curricula can lead to the same performance.
> 1.2) Despite that, when extrapolating the task to new settings, differences between networks trained with these curricula arise.
> 2) The source of these differences can be traced to dynamical objects formed through training - in our case slow points.
> 3) Analyzing these slow points in the nominal task can predict behavior on the extrapolated task
> 4) Tracing the formation of these slow points through learning provides a link between training, slow point formation, stability of memories and performance
>
> As for the toy setting - we opted for a setting that would allow us to parametrically extrapolate the task. Furthermore, we now expanded our results to another architecture (LSTM) and another dataset (CIFAR).
>
> We hope these changes amount to both clarifying exactly what question we are addressing, and broadening the results.

---

### Meta-Review · Area_Chair1 · 2018-12-12
**borderline**

**Confidence:** 2
**Recommendation:** Reject

**Metareview:**

This paper analyses the dynamics of RNNs, cq GRU and LSTM.

The paper is mostly experimental w.r.t. the difficulty of training RNNs; this is also caused by the fact that the theoretical foundations of the paper seem not to be solid enough.  Experimentation with CIFAR10 is not completely stable.

The review results make the paper balance at the middle.  The merit of the paper for the greater community is doubted, in its current form.